# Recent Advances on Nanofiber Fabrications: Unconventional State-of-the-Art Spinning Techniques

**DOI:** 10.3390/polym12061386

**Published:** 2020-06-20

**Authors:** Jinkyu Song, Myungwoong Kim, Hoik Lee

**Affiliations:** 1Division of Nano-Convergence Material Development, National Nano Fab Center (NNFC), Daejeon 34141, Korea; jksong@nnfc.re.kr; 2Department of Chemistry and Chemical Engineering, Inha University, Incheon 22212, Korea; 3Research Institute of Industrial Technology Convergence, Korea Institute of Industrial Technology, Gyeonggi-do, Ansan 15588, Korea

**Keywords:** nanofiber, electrospinning, needleless spinning, mass production, structural regulation

## Abstract

In this review, we describe recent relevant advances in the fabrication of polymeric nanofibers to address challenges in conventional approaches such as electrospinning, namely low throughput and productivity with low size uniformity, assembly with a regulated structure and even architecture, and location with desired alignments and orientations. The efforts discussed have mainly been devoted to realize novel apparatus designed to resolve individual issues that have arisen, i.e., eliminating ejection tips of spinnerets in a simple electrospinning system by effective control of an applied electric field and by using mechanical force, introducing a uniquely designed spinning apparatus including a solution ejection system and a collection system, and employing particular processes using a ferroelectric material and reactive precursors for atomic layer deposition. The impact of these advances to ultimately attain a fabrication technique to solve all the issues simultaneously is highlighted with regard to manufacturing high-quality nanofibers with high- throughput and eventually, practically implementing the nanofibers in cutting-edge applications on an industrial scale.

## 1. Introduction

Nanofibers belong to a group of most interesting and promising materials due to their unique physicochemical properties, i.e., extraordinary porosity with interconnectivity between pores in mats, mechanical flexibility and strength, high surface area, and high applicability to fabricate composites with other materials [1,2,3]. These characteristics enabling its use for a variety of applications have attracted tremendous scientific and technological interests [4,5,6,7,8]. Among various nanofiber fabrication techniques, such as melt spinning [9], wet spinning [10], dry spinning [11] and others, an electrospinning utilizing electrical force to form polymeric fibers with a diameter ranging from nanometer to sub-micrometer has been considered as an important workhorse academically as well as industrially. In the early days, Lord Rayleigh′s studies showed that an electrical force can overcome the surface tension of a small drop of solution [12]. While the studies have been an important background for developing electrospinning, Formhals registered a crucial patent related to electrospinning technique for the first time in 1934 [13]. In 1969, Taylor observed the formation of cone type jets from viscose fluids with an electrical field, the-so-called “Taylor cone” [14]. Since the 1980s up to now, the electrospinning technique has widely been utilized due to the accessibility and applicability to nanotechnology. More importantly, it has been rapidly evolved in its practical use on the basis of fundamental studies on significant aspects of the electrospinning process [15]. 

In general, electrospinning is regarded as an easily accessible, highly effective and efficient nanofiber fabrication method. One of its significant benefits is the versatility to use a number of types of materials, e.g., organic materials ranging from small molecules to macromolecules, inorganic species and particles, and organic/inorganic hybrids. A typical electrospinning apparatus consists of a high voltage power supply, a spinneret connected to the power supply, and a collector having the opposite charge to the ejection spinneret charge [16]. The fiber source is typically a homogeneous polymer solution which is injected into the spinneret. A polymer solution droplet at the end of a spinneret is exposed to an applied electric field, and the electric field induces charges on the droplet surface [17]. When the voltage is sufficiently high to overcome the droplet surface tension, the repulsive electrical force stretches out the droplet. The stretched-out jet is rapidly accelerated to form the structure of the nanofiber [2]. The morphology of the resulting nanofibers is effectively controlled by many specific processing parameters, i.e., solution viscosity, applied voltage, solution injection rate, distance between spinneret tip and collector, solvent conductivity, temperature, and humidity [18].

Despite the outstanding characteristics of electrospinning, improvement to address important issues is in progress by partially modifying the spinning process or even developing new types of spinning process and apparatus. Most notably, the throughput of conventional electrospinning has been a serious bottleneck in its practical applications. Furthermore, it is still challenging to achieve the fabrication of nanofibers with (i) a highly regulated structure, (ii) the desired fiber alignment and orientation, and (iii) three-dimensional complexity in the current electrospinning process. (Figure 1) In the perspective of the materials, there has been a significant limitation on the range of materials used in the fabrication; for example, some polymers, such as polyolefins, with poor solubility in a range of solvents, or with high electrical resistivity, cannot be utilized as a homogeneous polymer solution where reasonable electrical conductivity is required [19]. Considering these issues, much research effort has been devoted to achieve advanced spinning systems. To date, researches have focused on the invention of new spinneret systems with unique designs, e.g., blow-jet spinning [20], centrifugal spinning [21], microfluidic spinning [22], bubble electrospinning [23], core-shell electrospinning [24,25], and the combination of several current techniques. In this review, we explore several notable recent nanofiber fabrication techniques, and highlight novel and remarkable approaches which potentially offer more efficient and effective pathways towards high quality nanofiber structures in a rapid manner with regard to regulated structure and architecture with high complexity. We envision that strategies covered in the current review should have the potential to stimulate the development of other new techniques as well as to facilitate beneficial applications that have impact on our daily lives. 

## 2. Needleless Spinning for Large Scale Production 

In typical electrospinning to fabricate nanofiber mats, a single spinneret is employed because it provides a convenient and cost-effective process. However, it does not offer high throughput which is one of the important requirements for practical and industrial applications. The typical production rate of the single spinneret electrospinning falls in the range of 0.01–0.1 g·h^−1^ [26], which highly limits its applicability in practice. One of the simplest and easiest approaches to increase the throughput and enhance the productivity is the use of multiple nozzles [27,28]. However, there have still been challenges in multi-tip electrospinning; for example, undesirable interaction between formed jets which is caused by the interference of the electric fields, and difficulties in the cleaning process for a number of tips eventually degrades the quality of the fabricated nanofibers. In addition, clogging of the needle tip frequently occurs due to its small diameter, preventing continuous spinning. The multi-tip system also requires huge space for the equipment, leading to increase of cost of the production process. 

To overcome this issue, needleless electrospinning has been considered as an effective method to maximize the continuity of the fabrication process and therefore, the productivity. Contrary to conventional electrospinning, in the needleless electrospinning technique, an electrical force is applied on the liquid surface directly without using a needle nozzle [23,29]. To date, many needleless electrospinning techniques have been reported with systematic variation in the spinneret geometry, e.g., spiral coil [30,31], rotary cone [32], sprocket wheel disk [33], rotating disk [34], self-cleaning threaded rod [35], twisted wire [36,37], conical wire coil [38], metal dish [39], double-ring [40], curved slot [41,42], bowl [43], and tube [44]. The open spinneret in the needleless electrospinning suppresses the clogging of the tip, facilitating multiple jet operations. Consequently, far higher fabrication productivity is feasible: the quantitative range is from 0.5 g·h^−1^ to 600 g·h^−1^ which is much higher than that of conventional electrospinning with a single needle [45,46,47]. Representative improvements are summarized in Table 1. In this section, we describe the recently developed new techniques related to needleless electrospinning that can achieve an outstanding throughput. 

### 2.1. Double-Ring Slit as Electrospinning Spinneret

Contrary to conventional electrospinning which is a closed system, in a needleless spinneret, a polymer solution is exposed to air, inducing solvent vaporization during electrospinning and hence, resulting in inconsistency in the quality of fabricated fibers due to the changes in polymer solution concentration [23]. Recently, Wei et al. introduced a double-ring slit as a spinneret for electrospinning to address this challenge and to improve fabrication productivity by simultaneous formation of multiple jets during electrospinning [40]. The schematic diagrams for the double-ring slit needleless spinneret (DRSNS) setup are presented in Figure 2A,C, and the photographs from front and top views also are shown in Figure 2D,E. The double-ring based electrospinning setup is quite similar to conventional single needle spinneret (SNS) electrospinning; spinneret, syringe pump, high-voltage generator, and metal collector (Figure 2B); however, the key component for DRSNS based electrospinning is the use of a double-ring slit as spinneret. As shown in Figure 2C, the double-ring slit consists of four parts, i.e., outer ring, inner ring, inner core, and shell. The inner core and shell are made from polytetrafluoroethylene to insulate the current, but the outer and inner rings are composed of copper to effectively apply an electrical field to the polymer solution. The width of the inner and outer slit was 0.5 mm, and the diameter and height of the spinneret was 35 mm and 40 mm, respectively. The DRSNS was mounted on a pedestal, and the polymer solution was injected into the inner core of a double-ring spinneret, and the solution transported into the double-ring slit. The ejected polymer solution from the inner and outer circle slit is highly charged owing to the copper rings imparting an electrostatic repulsion force that counteracts the effect of surface tension. As a result, the polymer solution is allowed to be stretched out to form nanofibers. In this process, the loss of polymer solution is limited effectively, and the multiple formation of a Taylor cone of the polymer solution occurs and therefore, enhancing the mass production of nanofiber. The productivity of this technique was compared to the SNS, with experiments to fabricate poly(lactic acid) (PLLA) nanofibers. Noticeably, the productivity of the DRSNS-based electrospinning process was improved about 22 times, from 0.10 ± 0.03 g·h^−1^ to 2.25 ± 0.25 g·h^−1^ (Figure 2F), under the same spinning conditions including feed rate, voltage, and rotating speed. Two aspects made this improvement feasible: (i) precise control of the exposed area of the polymer solution to the atmosphere using a control pump which reduced the loss of polymer solution during the spinning process, and (ii) concurrent formation of multiple jets with the circular narrow slit. DRSNS was applicable to other types of polymers such as polyacrylonitrile, poly(vinyl alcohol), poly(methyl methacrylate), poly(DL-lactic acid), and polycaprolactone, to achieve a multilayered nanofiber structure which may be implemented further for bio-related applications such as drug loading systems.

### 2.2. Two-Level Coil Edge Electrospinning

Among a variety of possible spinneret structures, a curved geometry has attracted attention due to its high electrospinning process efficiency. Yan et al. investigated the effect of a slot line shape on electrospinning productivity and the quality of resulting nanofiber by comparing four convex slot spinnerets with different line shapes, i.e., straight, rectangular, triangular, and curved shapes [41]. In the curved slot, higher electric field with more uniform distribution could be applied along the slot length direction, resulting in structural uniformity as well as improved throughput. The profile of the formed electric field was critical; Wang et al. utilized a finite element method (FEM) to analyze the electric field intensity profiles of a cylinder, disc, and coil spinnerets [30]. They found that larger curvature leads to higher electric field on a coil surface. Maximizing this effect for needleless electrospinning was feasible with coil-based electrospinning. It typically uses a smooth wire coil as a spinneret [48]. However, there are also challenges such as corona discharge and breakdown of the electric field to make a curved structure with the spinneret [49]. Particularly, the electric field is often unevenly formed at a certain position in the curvature, inducing corona discharge which leads to unnecessary energy consumption and therefore, process efficiency becomes lower than expected. The additional use of energy also can result in unexpected equipment damage. Therefore, it is important to design a curvature shape of the spinneret to realize an efficient electrospinning system for ensuring the mass production of high-quality nanofibers with the desirable size uniformity.

Niu et al. recently presented a new type of coil-based electrospinning, which uses a secondary coil structure on the coil spinneret, to overcome the aforementioned challenges [31]. The two-level coil (T-Coil) electrospinning apparatus was built with a high voltage power supply, metal collector, helical coil, and plastic solution container, as shown in Figure 3a. The helical coil is partially immersed in the polymer solution bath, and it rotates along the axis during the spinning. The T-Coil spinneret is made by winding the second coil onto a normal coil (N-coil) with a spacer template to keep a consistent coil pitch. The electrical power is supplied by immersing an electrode into the plastic solution bath. With this setup, the effect of the secondary coil using FEM analysis was investigated, as shown in Figure 3b. Interestingly, the electric field in the T-Coil is predominantly centralized on the secondary coil rather than the primary coil, whereas that in the N-Coil is centralized on the primary coil. The secondary coil enhances the electric field on its surface, inducing a reduction of threshold voltage for jet initiation. In addition, the secondary coil provides an additional area for jet formation on its surface, resulting in the electrospinning productivity increasing up to 170% compared to the N-coil (N-coil productivity: 2.94 g·h^−1^, T-coil productivity: 8.1 g·h^−1^). These results highlight the importance of the design of a novel electrospinning spinneret with regard to mass production with high throughput. 

### 2.3. Rotary Cone as Electrospinning Spinneret 

Lu et al. demonstrated a super-high-throughput electrospinning technique employing an electriferous rotary cone as a spinneret [32]. This technique improved the productivity of nanofiber fabrication up to 600 g·h^−1^. This achievement is remarkable as the typical productivity of single nozzle electrospinning is in the range of 0.01–0.1 g·h^−1^ [26]. This electrospinning system has similar components to a typical electrospinning apparatus except for the cone-type spinneret. Upon the injection of polymer solution into the cone, a droplet is formed and charged immediately. The charged polymer droplet flows along the rotating surface of the cone under the coaction of gravity, the moment of inertia, and the electric force. Once the liquid droplet reaches the edge of the cone, it starts to deform and stretch out to the fiber structure. A number of jets are formed around the edge of the rotating cone as feeding of the polymer solution continues, resulting in an increase of production rate. The spinning rate is high enough to be applied to an industrial production process. It is notable that though this technique was reported in 2010, the productivity is the highest gain among nanofiber production techniques so far.

### 2.4. Foam Based Needleless Electrospinning 

Solution blow spinning which uses concentric nozzles and compressed gas also has attracted attention in nanofiber fields [50,51,52,53,54,55]. The high pressure gas such as air, nitrogen, and argon is released from the outer nozzle, and the polymer solution is ejected through the inner nozzle. This technique has also been considered as an effective way to achieve a higher production rate than that of typical electrospinning [55]. This process is kindred to the melt-blown technique. The polymer solution is stretched by the co-flowing gas jet from the outer jet, leading to formation of a polymer nanofiber structure. Similar to blow spinning, bubble spinning (which is named foam electrospinning) also uses a gas. However, the gas is injected with the submerged nozzles in solution making bubbles, leading to the possibility of fibers initiating from the bubble surface. This type of spinning using single or several bubbles in a vast bath of solution has been presented in various reports [56,57,58].

An interesting approach, by Hwang et al., to exploit the foam of polymer solution has recently exhibited improved productivity in Nafion nanofiber fabrication [45]. Nafion is a widely utilized polymer in the field of electrolyte membranes owing to its high water-saturated proton conductivity (0.1 S·cm^−1^), as well as its thermal, mechanical, and chemical stability [59]. Nafion has been engineered in many different physical forms, e.g., bulk such as pellet, film, and dispersion in a mixed solution of alcohol and water. Its nanofibrous structure has attracted much attention because it could improve proton conductivity and subsequently, the performance in certain applications [60,61]. In most studies, electrospun Nafion/polymer composite nanofibers formed by a single nozzle electrospinning have been demonstrated [62]. However, the fabricated Nafion typically exhibited low purity and as a result, low proton conductivity in its composite structure. Foam electrospinning addresses these issues by using compressed gas, e.g., CO_2_, to form polymeric foams where multiple polymer jets are ejected from the surface of the foam toward the collector. In the setup, a copper electrode is connected to a funnel to supply electrical power to the polymer solution (Figure 4a). The applied electric field disturbs the polymer solution foam formed by injected gas and as a consequence, multiple polymer jets are ejected toward the collection plate directly, as shown in Figure 4b. The multiple jets led to a high production rate of approximately 9 g·h^−1^·m^−2^ which is far higher than that of single nozzle electrospinning for Nafion nanofiber fabrication (typically 0.0014 g·h^−1^·m^−2^). In addition, the purity of Nafion nanofiber was about 98%, which has not been achieved with traditional electrospinning. The method was capable of resolving both productivity and purity issues in Nafion nanofiber fabrication.

## 3. The Use of Mechanical Force for Nanofiber Fabrication 

Despite the simplicity and versatility of electrospinning, it has intrinsic limitations for making nanofibers owing to the electrostatic repulsion force that counteracts the surface tension of the droplet. It has recently been demonstrated that the electrical force can be replaced with a mechanical force to fabricate nanofibers. The use of mechanical force overcomes the conventionally accepted limitations of electrospinning, e.g., requirement for high electrical potential and electrically conductive targets to drive the stretching-out of the droplets in the fiber elongation process. In contrast with electrospinning, the use of mechanical force does not limit the type of polymers and solvents because their electrical properties are no longer relevant. Furthermore, it avoids the excessive use of energy in production contributing to high cost. In this section, the efforts on using another type of force than electrical force to make high-quality nanofiber are discussed.

### 3.1. Handspinning

As an alternative to electrical force, Lee et al. developed the-so-called handspinning method [63]. It only relies on simple mechanical stretching forces instead of high electrical voltage during the fabrication process of nanofiber, which is advantageous in reducing the high cost as well as excessive energy use for fiber production. Figure 5 shows the setup of the handspinning apparatus. Nanofibers are formed by simple attachment–detachment of two plates where polymer solution is sandwiched by the plates, resulting in well-oriented nanofiber along a single axis. The nanofiber formation can be controlled by varying the process parameters of pulling-away speed, pulling-away distance, and plate area. The unique approach using the uniaxial stretching-out force offers a handle to control the internal structure of nanofiber composites: carbon nanotube (CNT) was evenly distributed in a polymer fibrous matrix minimizing its agglomeration, which is commonly observed in electrospinning. Furthermore, CNTs are aligned along the direction of the pulling-out motion, which is quite challenging to achieve with electrical force. Consequently, a controlled structure was integrated for the enhancement of mechanical and thermal properties of the nanofiber composites. These observations highlight the potential of mechanical force to fabricate a nanofibrous structure with controlled complexities which can be directly related to the functional materials platform.

### 3.2. Needle Spinning

Although handspinning offers many advantages, unevenly supplied polymer solution on pulling plates leads to low size uniformity in the resulting nanofibers. This issue was addressed by changing the plate to needle as a tool for drawing the polymer solution, the-so-called needle spinning [64]. Needle spinning is capable of achieving highly uniform nanofibers by supplying a certain amount of polymer solution by generating a uniform meniscus on the needle tips. The process is illustrated with a photograph of a home-built apparatus in Figure 6. By simply dipping a needle in the polymer solution and drawing out the solution, a single nanofiber is readily fabricated. On multiple needles, a highly regular shape of the meniscus at the tips of the different needles is realized. Processing parameters, i.e., the pulling-away speed, pulling-away distance, needle size, and polymer concentration were effective in controlling the size of the resulting nanofiber. Clear correlation between the nanofiber diameter and processing parameters was observed: the nanofiber becomes thicker as both polymer concentration and needle size increase, and it becomes thinner as both pulling-away speed and pulling-away distance increase. The needle-spun nanofibers exhibited a smooth surface and uniformly controlled diameter as shown in Figure 6. The length of nanofiber can be controlled up to the meter scale, which is also a remarkable improvement of mechanical force-based spinning techniques. These studies elucidate the fundamental principles for a mechanically driven nanofiber formation process, and show an effective way to produce well-defined nanofiber with the desired dimension. 

### 3.3. Track Spinning

The drawing-based technique has been further improved towards a continuous fabrication process using an automated single-step drawing system [65]. The process, called track spinning, is based on a simple drawing method using two oppositely angled rotating tracks. The equipment consists of two rotating tracks touching each other at the top, a solution dispenser located above the tracks, and a collection rack located below the tracks (Figure 7a). Similar to handspinning and needle spinning, the driving force to form nanofibers is a mechanical force, which allows a range of options for the selection of polymer and solvent. Upon feeding the polymer solution by dispenser at the top of the apparatus, fibers are continuously spun by mechanical drawing with the increase of the distance between the tracks due to the geometry of the apparatus. The polymer solution is coated uniformly over the surface of two tracks at the initial stage, and it is steadily stretched out across the gap between the tracks around the third–fourth stage. Pulled-out fibers are subsequently solidified by solvent evaporation as the fibers move down along the track. As this process is repeated by continuous rotation of the tracks, a large amount of fibers can be fabricated. The fabricated nanofibers are collected onto the collection frame at the bottom. The size of the nanofibers is effectively controlled in a nanometer scale with the processing parameters of the angle of the track, the vertical collection distance, and the track speed. This approach maximizes the use of mechanical force capable of highly aligning nanofibers with high throughput owing to its continuity. Furthermore, the setup can be easily modified: for example, the size and texture of tracks improve the production rate and the feasibility to control the location of drawing and therefore, the location of fibers (Figure 7d). As a proof of concept, poly(vinyl acetate) and polyurethane nanofibers were fabricated with a diameter of about 500 nm and the length of 255 mm. Therefore, spinning with a mechanical force clearly proves the feasibility of mass production of well-defined nanofibers, which have an impact in the field of nanofiber applications.

## 4. Structural and Architectural Regulations 

Most researches conducted with typical electrospinning have demonstrated randomly oriented nanofibers exhibiting a circular cross-section and smooth surface. To expand the applicability of nanofibers, however, constructing particular architectures in two and even three dimensions with structurally well-defined nanofibers is critical. For example, nanofiber patterns with controlled alignment on a two- or three-dimensional substrate were demonstrated for certain purposes [66,67,68,69]. Specific surface morphologies with highly regulated fiber structures are often desirable [68,70,71,72,73,74]. In this section, recent unique demonstrations on controlling nanofiber morphology, alignment, pattern formation, and constructing a three-dimensional structure with electrospinning are introduced.

### 4.1. Bipolar Pyroelectrospinning for the Formation of Nanofiber Arrays 

As aforementioned, electrospun nanofiber is generally collected as a nonwoven type due to the uncontrollable nature of jet formation, which limits its applicability. Enormous interest in controlling the nanofiber deposition process has arisen to improve the performance in specific applications. A number of studies dealt with patterning techniques with electrospun nanofiber, including precise positioning, stacking, and orientation [75,76], selective deposition [77], and templated patterning [78]. These techniques provide significant impacts especially in the field of biomedical engineering, such as tissue engineering [79] and regenerative medicine [80]. These efforts have evolved to find an easy and efficient method to achieve the desired pattern formation using electrospun nanofibers in high precision. 

Recently, Grilli et al. invented bipolar pyroelectrospinning, which is capable of generating ordered arrays of nanofibers by exploiting periodically poled lithium niobate (PPLN) as a substrate [81]. First, PPLN crystal was patterned in a hexagonal shape, where ferroelectric domains are formed with reversed polarization, achieved by a standard electric field poling onto a template sample defined with photolithography. The optical microscopic image of hexagonal PPLN crystal pattern is shown in Figure 8a. Heat is an important stimulus for this pattern: charges are homogeneously distributed on the surface having the PPLN crystal patterns when the heating is off. In this state, only the crystal surface exposed to the c ends has a spontaneous polarization state. On the opposite side from the PPLN crystal side, a drop of polymer solution is formed on the metallic tip, similar to the typical electrospinning process. When the heat is switched on to render the plate at a temperature of approximately 60 °C, the charge state of the PPLN crystal pattern becomes different, resulting in a heat-induced pyroelectric effect. As a result, the regions near the hexagons experience reversed polarization to exhibit an excess of negative charges, and the hexagons exhibit an excess of positive charge. This bipolar electric field patterns guide the location of the deposited nanofibers during electrospinning. If the ejected nanofibers are positively charged, they effectively interact with the negatively charged area outside of the hexagons and therefore, patterned nanofiber mat is formed. (Figure 8d). The resulting patterns prove the validity of this technique to realize periodically aligned arrays. 

### 4.2. Near-Field Electrospinning to Construct Nanoarchitecture

The electrospun jet is highly uncontrollable due to the chaotic whipping jet formation, which makes a randomly orientated nonwoven type nanofiber mat. To suppress the chaotic whipping jet formation and to deposit nanofibers on a substrate with a desired pattern, the near-field electrospinning (NFES) technique has been developed in recent years. NFES provides a straightforward and versatile means to precisely control the location of nanofiber and therefore, to fabricate nanofiber patterns. It features a short tip-to-collector distance less than 3 mm, which leads to a reduction of bending instability of the electrospun jet, and a lower voltage of several hundreds of volts is required. The first proof-of-concept was demonstrated by Lin et al. in 2006 [82]. The NFES technique requires a relatively low voltage for processing, and enables the fabrication of pattern arrays with excellent position control, while excessive consumption of polymer solution can be minimized. Initially, it was employed to attain only 2D structures; however, recently building a 3D structure has been extensively undertaken. The most use of this method is for the layer-by-layer additive printing method [83]. Hutmacher et al. showed highly ordered scaffold architecture using multilayer additive printing by spinning similar to NFES with highly viscous polymer solution [84]. As a result, fibers with several micrometer dimension were stacked to form the scaffold. Three dimensional architecture in nanometer dimension was presented by Lee et al. in 2014 [85]. In this work, a pre-defined pattern on the conducting electrode guides the location of spun nanofibers, which could however limit the expansion of its applicability. Another 3D architecture, a hollow pottery shape, was fabricated by Kim et al. using the spontaneous coiling of electrospun nanofiber, demonstrating a hollow cylindrical structure [86]. The a layer-by-layer additive printing method with the NFES was successful in achieving the desired shapes, though a pre-defined structure or specifically a pre-designed collector is required. 

Recently, Cho et al. successfully fabricated nanofiber stacks built with high-resolution control using the NFES technique [87] (Figure 9a). In their method, a high aspect ratio in a variety of architectures such as curved nanowall, grid pattern, and nanobridge, was achieved without any pre- designed collector because of the self-alignment behavior of the nanofibers. The self-alignment feature was possible by the simple addition of salt, i.e., poly(ethylene oxide) (PEO)/sodium chloride (NaCl) aqueous solution. When pure PEO solution is electrospun, the charge does not fully dissipate, leaving a weak positive charge on nanofiber surface. The weak positive charge induces the repulsive interaction in jet streams and as a consequence, stacking of nanofibers becomes challenging. On the other hand, additional salt enhances the conductivity of the polymer solution, and the surface of the resultant deposited nanofibers becomes negatively charged. The increase of the interaction between polymer solution jet and deposited fibers leads to effective stacking of the nanofibers (Figure 9b). This approach facilitated the fabrication of various nanoarchitectures including nanowalls, curved nanowalls, grids, and nanobridges at the desired locations with pre-programmed X-Y stage motion of the spinneret, and subsequent metal coating to enhance mechanical stability (Figure 9c).

### 4.3. Sequential Metal Deposition for Multilayered Nanofiber

Cross-section morphology control of nanofibers, e.g., fabrication of core-shell type or multilayered nanofiber which can be utilized for a variety of applications, has been an attractive but challenging issue [88,89,90]. Recently, an interesting facile fabrication approach for multilayered nanofiber was demonstrated by Aydin et al. [91]. They fabricated strong, flexible, and centimeter- long nanowires through multiple metal depositions on a polymeric nanofiber, as shown in Figure 10. Unlike a conventional core-shell fiber formation method which is enabled by ejecting two polymer solutions into one spinning tip, this method involves a physical vapor deposition (PVD), chemical vapor deposition (CVD), and atomic layer deposition (ALD). First, electrospun nanofibers of poly(*m*-phenylene isophthalamide) (PMIA) were collected on a spoked-drum collector for fabricating the aligned and free-standing nanofibers. (Figure 10a) Then, the fibers were transferred onto a rectangular frame to suspend them. Subsequently, the fixed nanofibers were coated with various functional metals or oxides using CVD, ALD, and PVD. Fabricated nanofibers exhibited various cross-section images, i.e., round-, oval-, and ribbon-shaped, controlled by the chamber humidity. (Figure 10b) This is due to the poor solubility of PMIA in water inducing a phase separation in the jet when water molecules from the atmosphere diffuse into the spinning jet. The number of layers were controlled by the number of deposition cycles, which is clearly assured in Figure 10b. This method opens up the possibility to finely control the internal structure of nanofibers, and therefore, eventually to achieve high structural complexities in nanofibers. 

### 4.4. Transformation of 2D Nanofiber Mat to 3D Object 

Nanofiber-based 3D scaffolds have been considered as a material platform in tissue engineering as they can be implemented to mimic the extracellular matrix (ECM) in native tissues. However, despite the development of various useful fabrication techniques, building a three-dimensional structure with nanofibers still remains a tough challenge. Widely utilized methods for transforming a 2D structure into a 3D object are the folding, rolling, bending, cutting, and buckling of 2D objects [92,93,94]. Recently, Jiang et al. reported the establishment of a 3D structure from 2D nanofiber mats of poly(ε-caprolactone) using a gas flow to expand electrospun nanofiber along the fiber deposition direction [95,96,97]. With this technique, the thickness and porosity in the highly ordered scaffold architecture were controlled. These nanofiber-based 3D architectures consist of multilayers of aligned nanofibers where the layers have gaps ranging from several micrometers to millimeters. Fabricated scaffolds have shown the feasibility for utilization in tissue engineering [96]. More recently, the gas flow technique was further developed to achieve a hierarchical assembly of nanofibers with pre-designed shapes having high complexity [98]. The key process of the gas-assisted expansion technique is to fix one side of the 2D nanofiber mat during the process, as presented in Figure 11a. Both processes include thermal treatment at 85 °C for 1 s before the expansion step to selectively harden one side of the nanofiber mat. Depending on the cutting shape of the 2D nanofiber mat, i.e., rectangle, triangle, half circle, and arch, and on the position of thermal treatment, the 2D nanofiber mat was transformed into a cylinder, cone, sphere, and hollow sphere, respectively, as shown in Figure 11b. Furthermore, changing the axis for rotation during the process allowed the fiber alignment direction in the resulting 3D object to be controlled. In the resulting materials, shape recovery upon compression was possible, and its porous and layered structure effectively guided the organization of seeded cells, enabling a highly ordered artificial tissue. 

### 4.5. Yarn-Spinning

Although electrospun nanofibers have various unique and valuable properties, they are typically known not to be sufficiently mechanically robust to replace common textiles. A few reports have shown that individual electrospun nanofibers can exhibit a reasonable mechanical strength despite their size in nanoscale [1,99,100], however, the final strength of the nanofiber mat is not enough to be utilized practically. For actual applications, nanofiber mats need to be supported by nonwovens or other substrates. Recently, Kim et al. demonstrated a yarn spinning technique to fabricate a nanofiber based single-stand yarn by employing a dual-spinneret electrospinning process on a support wire (Figure 12a) [101]. They used a funnel as a collector to wind the electrospun nanofiber onto the support wire, and the dual spinning nozzle for electrospinning. Simultaneously, the electrospun nanofibers were collected as a yarn on the winder like a thread. The fabricated nanofiber yarn has a core-shell type structure where the core is a conventional thread yarn, and the shell consists of electrospun nanofibers with high density. In this technique, it is important to control the surface charge on the polymer droplets formed by electrospinning, as positively and negatively charged electrospun nanofibers are intermixed on the thread which is located in the funnel. This flexible yarn platform was further implemented for hydrogen sensing by coating the yarn with sequential sputter depositions of Pd and Pt. The fabricated metal-polymer composite single-strand yarn showed a wide detection range from 4 to 0.0001% with long-term stability toward repeated exposure of high-concentration H_2_ (4%). It also exhibited dramatically rapid sensing speed and strong mechanical bending strength, ensuring the potential as a usable sensor platform. This approach is not complicated for fabricating a flexible and free-standing single-stand yarn scaffold with high-density polymer nanofibers, but it provides different functional core-shell (thread-nanofiber yarn) type fabrics with high flexibility, high surface area, and open porosity.

## 5. Closing Remarks and Outlook

Nanofibers have attracted tremendous technological interest, and related researches are rapidly moving forward to exploit significant characteristics of nanofibers such as high porosity with excellent pore interconnectivity between pores in mats, flexibility with reasonable strength, high surface-to-mass ratio, and the ability to incorporate other materials into actual applications. However, their use in industries have been still challenging due to the poor throughput and productivity of the current fabrication processes, and the difficulties in regulating the structure, alignment, and orientation. In the productivity perspective, electrospinning, one of the popular fabrication techniques, exhibits productivity in the range of 0.01–0.1 g·h^−1^, and the principle for fiber formation does not allow the realization of complex nanofiber structures or patterns. However, other approaches to resolve these issues have recently been proposed. The most important issue is the increase of productivity. The use of multiple nozzles can be the easiest way to do so; however, it brings several issues for processing, and requires an increase of apparatus size, which also increases the cost of equipment as well as of the space. As an alternative, the use of an open spinneret system such as needleless spinning has been shown. This system allows direct application of an electrical force to the liquid surface, generating multiple jets and eventually, achieving an impact on productivity with up to 600 g·h^−1^. However, it is typically conducted in an open system with a complicated setup requiring the control of many different processing parameters. Consequently, the consistency of the nanofiber morphologies is largely affected, and the complexity in the system negates its applicability in industry. Another issue to have arisen in the spinning technique is the regulation of structure regulation with the control of alignment and orientation. Uniquely designed, but complicated processes and specially outfitted apparatus for patterning or constructing specific structure of nanofibers have been proposed; however, they still suffer from low throughput. Finding an innovative and creative fabrication technique is still ongoing in order to solve all the issues simultaneously to realize high-throughput and easy operation and eventually, to bring the use of nanofibers to the real world. We anticipate that the currently devoted efforts by a number of researchers will eventually bring advanced technologies into this field, with regard to practical implementation in industrial applications. 

## Figures and Tables

**Figure 1 polymers-12-01386-f001:**
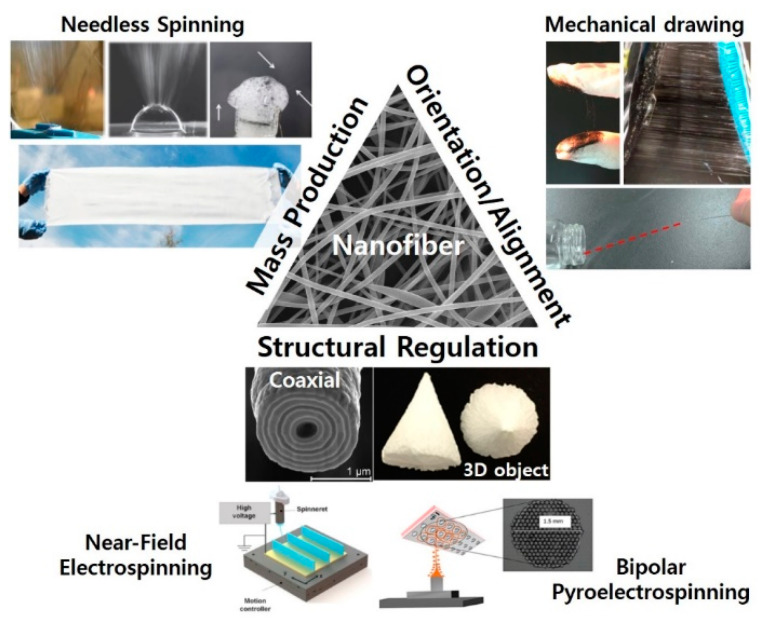
Challenges in the conventional nanofiber fabrication technique and different notable approaches for advanced nanofiber production. [Reproduced with permission from 42, 63, 98, 81, 87, and 91. Copyright (2017, 2016, 2019, 2019, 2019 and 2019) American Chemical Society].

**Figure 2 polymers-12-01386-f002:**
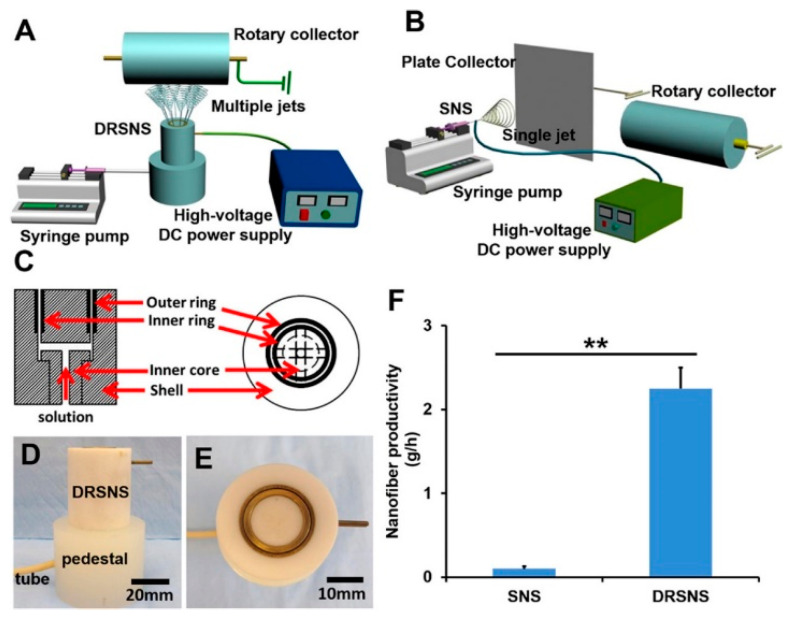
Schematic diagrams of (**A**) the double-ring slit needleless spinneret (DRSNS) setup, (**B**) single needle electrospinning setup, and (**C**) diagram of the internal structure of the DRSNS apparatus, photographs of DRSNS from (**D**) front view and (**E**) top view, and (**F**) a plot showing comparison of nanofiber productivities of DRSNS- and single needle spinneret (SNS)-based electrospinning systems. [Adapted with permission from 40. Copyright (2019) American Chemical Society].

**Figure 3 polymers-12-01386-f003:**
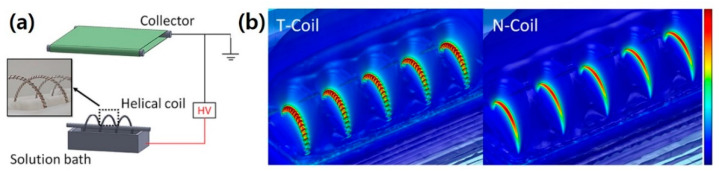
(**a**) Schematic illustration of the two-level coil electrospinning setup, and (**b**) electric field intensity profiles of two-level coil (T-Coil) and normal coil (N-Coil). [Adapted with permission from 31. Copyright (2018) American Chemical Society].

**Figure 4 polymers-12-01386-f004:**
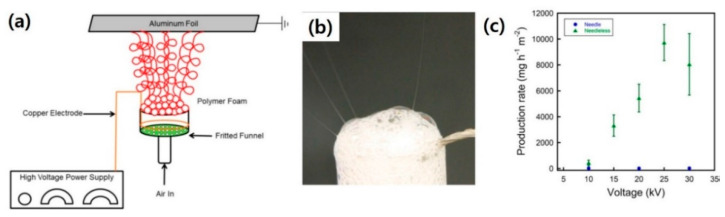
(**a**) Schematic illustration of foam based electrospinning, (**b**) photograph of fiber generation from the foam surface, and (**c**) Nafion nanofiber production rate of traditional electrospinning and foam based needleless electrospinning. [Adapted with permission from 45. Copyright (2019) American Chemical Society].

**Figure 5 polymers-12-01386-f005:**
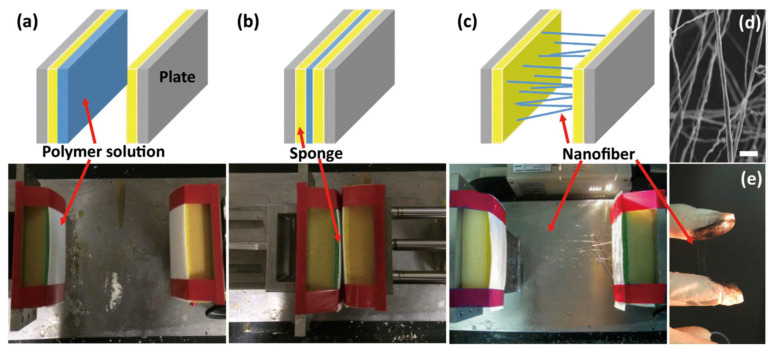
(**a**–**c**) The nanofiber fabrication process via handspinning, (**d**) representative SEM image of handspun nanofibers (scale bar = 10 μm), and (**e**) photograph showing handspun CNT-reinforced nanofibers using two fingers. [Adapted with permission from 63. Copyright (2016) Nature Publishing Group].

**Figure 6 polymers-12-01386-f006:**
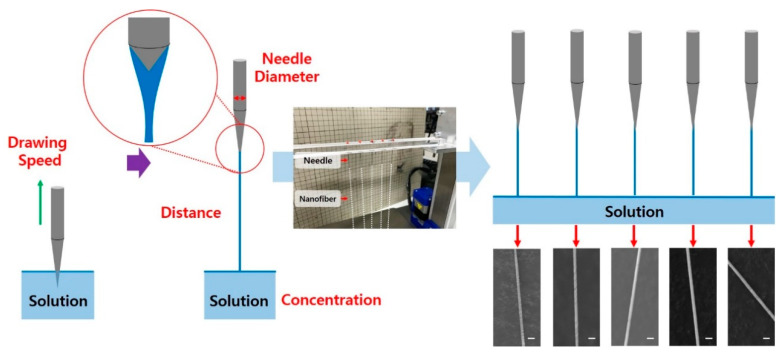
The needle spinning apparatus, and SEM images of nanofibers fabricated from needles at different locations in the same batch (scale bar = 2 μm). [Adapted with permission from 64. Copyright (2018) MDPI].

**Figure 7 polymers-12-01386-f007:**
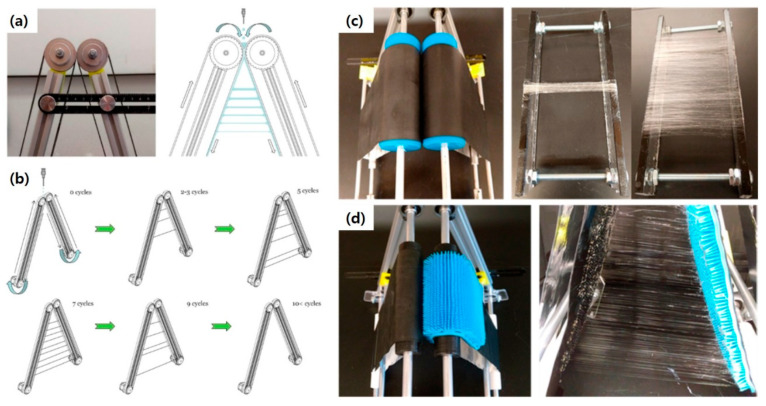
(**a**) The setup for track spinning, (**b**) schematic illustration of continuous fiber formation, (**c**) photographs of track spinning process with 8 cm wide tracks and of resulting aligned fibers, and (**d**) photograph of a 3D spinning system with a patterned array of bristles on one track towards precise location of fibers. [Adapted with permission from 65. Copyright (2019) American Chemical Society].

**Figure 8 polymers-12-01386-f008:**
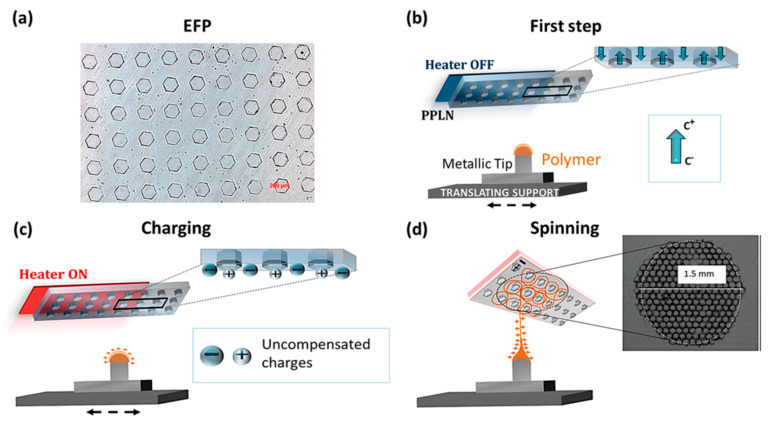
Schematic illustration of the bipolar pyroelectrospinning process. (**a**) Optical microscope image of the periodically poled lithium niobate (PPLN) crystal plate, (**b**) the pyroelectrospinning with heat-off and (**c**) with heat-on where the PPLN plate becomes charged. The polymer droplets exhibit positive charges to deform into Taylor cone. (**d**) The ejected nanofibers collected on the region surrounding the hexagons, and the SEM image of the resulting pattern array in the nanofiber mat. [Adapted with permission from 81. Copyright (2019) American Chemical Society].

**Figure 9 polymers-12-01386-f009:**
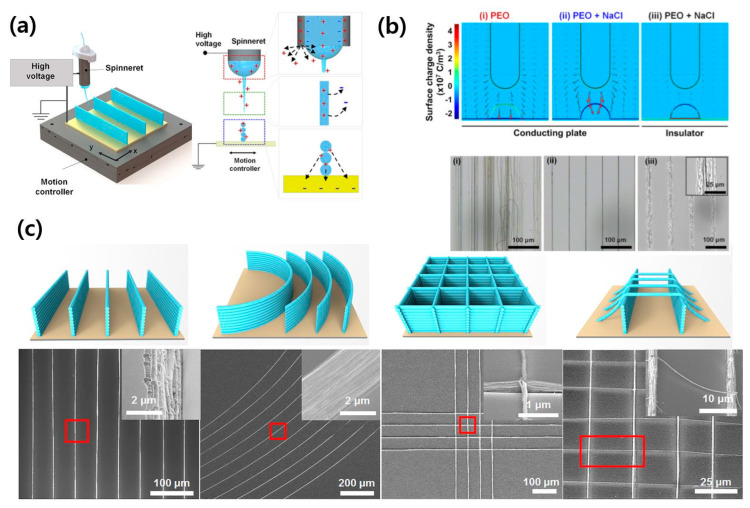
(**a**) Schematic illustration of experimental setup of NFSE, (**b**) diagram showing surface charge density for pure poly(ethylene oxide) (PEO) and PEO/NaCl on conducting plate, and PEO/NaCl on insulating plate, and (**c**) fabricated various 3D nanoarchitecutres, i.e., nanowalls, curved nanowalls, grid pattern, and nanobridges. [Adapted with permission from 87. Copyright (2019) American Chemical Society].

**Figure 10 polymers-12-01386-f010:**
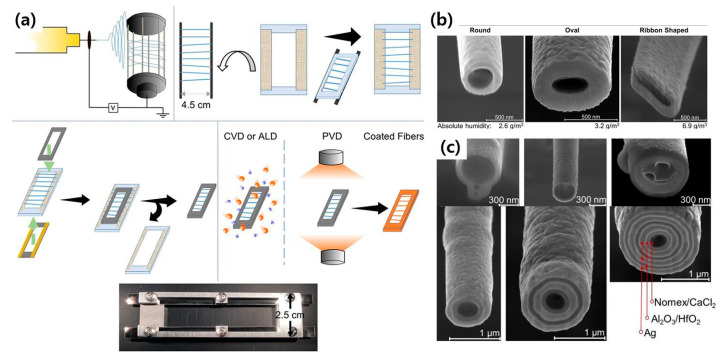
(**a**) Schematic illustration of the process for multi-layered fibrous structures, (**b**) cross-sectional SEM images of of poly(*m*-phenylene isophthalamide (PMIA) nanofibers with various cross-sectional structure by humidity control, and (**c**) layer-by-layer metal deposited PMIA nanofibers via atomic layer deposition (ALD) and chemical vapor deposition (CVD). [Adapted with permission from 91. Copyright (2019) American Chemical Society].

**Figure 11 polymers-12-01386-f011:**
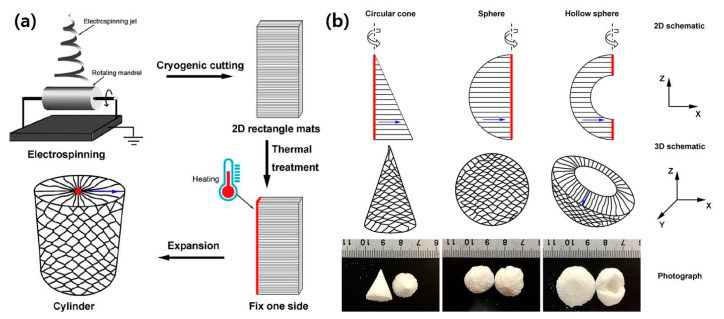
(**a**) Representative process for transformation of 2D nanofiber mats into 3D object, and (**b**) design and realization of the desired shapes. [Adapted with permission from 98. Copyright (2019) American Chemical Society].

**Figure 12 polymers-12-01386-f012:**
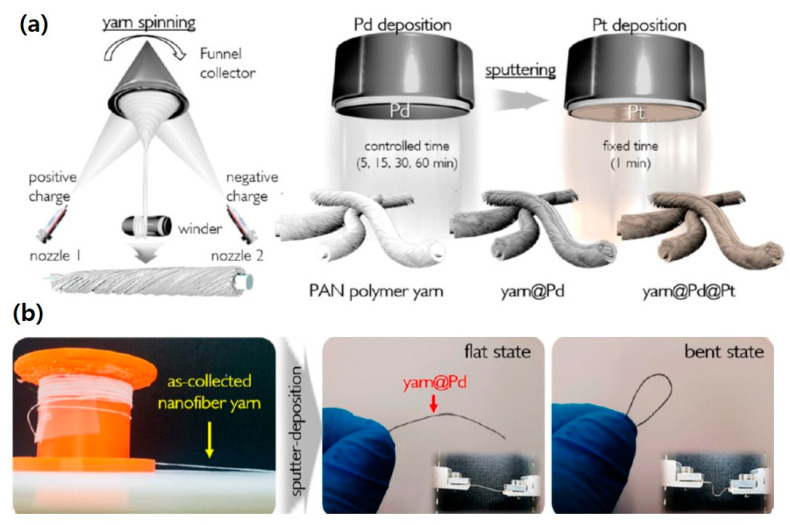
(**a**) Schematic illustration of yarn spinning followed by a metal deposition process, and (**b**) photographs of as-fabricated nanofiber yarn and Pd-coated yarn. [Adapted with permission from 101. Copyright (2019) American Chemical Society].

**Table 1 polymers-12-01386-t001:** Summary of productivity of different spinning techniques.

Type	Spinneret	Polymer	Voltage (kV)	Productivity (g·h^−1^)	Ref
Needleless electrospinning	Secondary coil (T-coil)	PVA	85	~9.72	[31]
	Double-ring slit	PVA	30	~2.25	[40]
	Twisted wire	PVP	20	~1.023	[37]
	Rotating-disk	PCL	25	~10.611	[34]
	Threaded rod	PEO	60	~5.2	[35]
	Rotating spiral wire coil	PVA	60	~9.42	[30]
	Curved convex slot	PVA	70	~2	[41]
	Foam	Nafion	25	~9.73	[45]
	Bowl edge	PEO	16	~0.684	[43]
	Rotating cone	PVP	30	~600	[32]
	Umbrella nozzle	PLLA	30	~180	[46]
	Curved slot with temperature elevation	PVA	60	~1.98	[42]
	Needle Roller	PVA	40	12.8	[47]
Nozzle electrospinning	Multi-nozzle (19 nozzle)	PEO	15	~0.712	[27]
	Coaxial with air-blowing	PAN(Core)/TPU(Shell)	38	~3.6	[25]
	Porous hollow tube (13 cm long, 20 holes)	PVP	40–60	0.3–0.5	[28]

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
