# Peer review of "Recent Advances on Nanofiber Fabrications: Unconventional State-of-the-Art Spinning Techniques"

_polymers, 2020, doi:10.3390/polym12061386_

Round 1

Reviewer 1 Report

This review manuscript offers one comprehensive summary of recent advances in nanofiber fabrication, from the novel spinning techniques, regulation of nanofiber structure and its organization, to key issues of nanofiber fabrication development. It is suggested to be accepted after minor modification. Some suggestion and comments are listed below.

1) Nanofibers are not commonly porous, but their mats or gatherings are porous. Thus, the statements like in Line 32, Line 497 and 498 are suggested to narrate in more exact way.

2) Line 58: “solvent viscosity” should be “solution viscosity”.

3) The description in Line 80-87 might not be included by this review.

4) Section 2 and Section 4 are concerned with the fabrication techniques, which are suggested to appear sequentially.

5) Line 115: “an order” is not exact, considering the difference between “0.5 g/hr to 600 g/hr” and “of 0.01-0.1 g/hr” (Line 97).

6) Type-errors: “from to fiber structure” (Line 208); “g/h·m2” (Line 230 and 231); “The a layer-by-layer” (Line 306). Pat attention to the subscript and superscript of “H2” (Line 397) and “ºC” (Line 367).

7) Based on SEM images of Figure 10 and Figure 11, the fiber diameter is about one micrometer, not in the nano-scale.

Author Response

Reviewer 1

This review manuscript offers one comprehensive summary of recent advances in nanofiber fabrication, from the novel spinning techniques, regulation of nanofiber structure and its organization, to key issues of nanofiber fabrication development. It is suggested to be accepted after minor modification. Some suggestion and comments are listed below.

 1) Nanofibers are not commonly porous, but their mats or gatherings are porous. Thus, the statements like in Line 32, Line 497 and 498 are suggested to narrate in more exact way.

Response:

Thank you for the comment. The comment is reflected in the main text to express the exact meaning.

(Line 32, Line 496)

2) Line 58: “solvent viscosity” should be “solution viscosity”.

Response:

Thank you for the comment. We have now reflected the reviewer’s point. (Line 58)

3) The description in Line 80-87 might not be included by this review.

 Response:

Thank you for the comment. We have now reflected the reviewer’s point by removing the paragraph.

4) Section 2 and Section 4 are concerned with the fabrication techniques, which are suggested to appear sequentially.

 Response:

Thank you for the comment. The Section 4 was moved to after Section 2 as Section 3 as the reviewer suggested.

5) Line 115: “an order” is not exact, considering the difference between “0.5 g/hr to 600 g/hr” and “of 0.01-0.1 g/hr” (Line 97).

 Response:

Thank you for the comment. The expression “an order of magnitude” was changed to “much”. (Line 106)

6) Type-errors: “from to fiber structure” (Line 208); “g/h·m2” (Line 230 and 231); “The a layer-by-layer” (Line 306). Pat attention to the subscript and superscript of “H2” (Line 397) and “ºC” (Line 367).

Response:

Thank you for the comment. The expression “from to fiber structure” was changed to “from the fiber structure”. (Line 198); the unit “g/h·m2” was fixed as “g·h-1·m-2. (Line 233, 235) ; and all the subscripts and superscripts were fixed. (Line 460, 490)

7) Based on SEM images of Figure 10 and Figure 11, the fiber diameter is about one micrometer, not in the nano-scale.

Response:

The presented fibers in Figure 10 were fabricated by two fingers, not handspinning apparatus. Although the fiber diameter in Figure 10 has micrometer scale, the dimension of the fabricated fibers by handspinning apparatus can be easily reduced to nanoscale. We show other SEM images below, which extracted from reference 53. The images clearly show nanofibers in SEM and TEM images.

We note that according to reference 54, the diameters of fibers in Figure 11 are 493, 494, 499, 501, and 490 nm, respectively (from left to right). These values are in typical length scale that is observed in nanofiber fabrications.

Reviewer 2 Report

The detailed review on mass production, orientation/alignment and structural regulation of electrospinning nanofibers was given, it would be better to give a detailed review on the new area of applications, such as liquid crystal encapsulation of electrospinning fibers (Guan et al., Colloids and Surfaces A 546 (2018) 212–220)).

Author Response

Reviewer 2

The detailed review on mass production, orientation/alignment and structural regulation of electrospinning nanofibers was given, it would be better to give a detailed review on the new area of applications, such as liquid crystal encapsulation of electrospinning fibers (Guan et al., Colloids and Surfaces A 546 (2018) 212–220)).

Response:

Thank you for comment. However, the reviews on the applications of nanofibers such as lithium-ion batteries, tissue engineering, scaffolds, and so on, can be easily found in literature. Typical examples are below:

  1. Electrospun Nanofiber-Based Anodes, Cathodes, and Separators for Advanced Lithium-Ion Batteries (Polymer Reviews, 2011, 51, 239.)
  2. Nanofiber technology: Designing the next generation of tissue engineering scaffolds (Advanced Drug Delivery Reviews, 2007, 59, 1413.)
  3. Co‐axial Electrospinning for Nanofiber Structures: Preparation and Applications (Polymer Reviews, 2007, 48, 353.)
  4. Electrospinning Cellulose and Cellulose Derivatives (Polymer Reviews, 2008, 48, 378.)

In current manuscript, we aim to focus on new nanofiber fabrication methods rather than applications. The advances in fabrication method are essential towards further improvements in different application fields. As a consequence, the reviews regarding nanofiber applications are not deeply discussed in this review.

However, since the recommend article is a nice and worthy reference in our review, we cited it as reference 8:

  1. Guan, Y.; Zhang, L.; Li, M.; West, J.L.; Fu, S. Preparation of temperature-response fibers with cholesteric liquid crystal dispersion. Colloids and Surfaces A: Physicochemical and Engineering Aspects 2018, 546, 212-220.

Reviewer 3 Report

In this paper, the authors reviewed unconventional spinning process for making nanofibers. However, the authors seem to mostly discuss about electrospinning and various ways to make nano fiber using an electricity assisted nanofiber making process. However, there are some significant work done on making nanofibers using compressed air- namely solution blow spinning/ solution blowing. The authors need to describe about them too. Some key work in that fields are, which the authors should refer to

(a)Solution blow spinning: A new method to produce micro‐ and nanofibers from polymer solutions (https://doi.org/10.1002/app.30275).

(b) Nano and submicrometric fibers of poly(D ,L ‐lactide) obtained by solution blow spinning: Process and solution variables (https://doi.org/10.1002/app.34410)

(c) Production of submicrometric fibers of mullite by solution blow spinning (SBS)(https://doi.org/10.1016/j.matlet.2015.02.111)

(d) Solution Blowing of Soy Protein Fibers (https://doi.org/10.1021/bm200438v)

(e) The production of 100/400 nm inner/outer diameter carbon tubes by solution blowing and carbonization of core–shell nanofibers (https://doi.org/10.1016/j.carbon.2010.05.056)

(f) Theoretical and experimental investigation of physical mechanisms responsible for polymer nanofiber formation in solution blowing (https://doi.org/10.1016/j.polymer.2014.11.019)

(g) Spray in Polymer Processing (https://doi.org/10.1007/978-981-10-7233-8_3) 

Author Response

In this paper, the authors reviewed unconventional spinning process for making nanofibers. However, the authors seem to mostly discuss about electrospinning and various ways to make nano fiber using an electricity assisted nanofiber making process. However, there are some significant work done on making nanofibers using compressed air- namely solution blow spinning/ solution blowing. The authors need to describe about them too. Some key work in that fields are, which the authors should refer to

(a) Solution blow spinning: A new method to produce micro‐ and nanofibers from polymer solutions (https://doi.org/10.1002/app.30275).

(b) Nano and submicrometric fibers of poly(D ,L ‐lactide) obtained by solution blow spinning: Process and solution variables (https://doi.org/10.1002/app.34410)

(c) Production of submicrometric fibers of mullite by solution blow spinning (SBS)(https://doi.org/10.1016/j.matlet.2015.02.111)

(d) Solution Blowing of Soy Protein Fibers (https://doi.org/10.1021/bm200438v)

(e) The production of 100/400 nm inner/outer diameter carbon tubes by solution blowing and carbonization of core–shell nanofibers (https://doi.org/10.1016/j.carbon.2010.05.056)

(f) Theoretical and experimental investigation of physical mechanisms responsible for polymer nanofiber formation in solution blowing (https://doi.org/10.1016/j.polymer.2014.11.019)

(g) Spray in Polymer Processing (https://doi.org/10.1007/978-981-10-7233-8_3) 

Response:

Thank you for the suggestion. We agree with the reviewer’s point that the solution blow spinning contributes much in nanofiber research fields. However, our review pursues to summarize the recent unconventional spinning process reported in very recent years, approximately in 2018-2020. The solution blow spinning is widely utilized and well-known process in fiber researches, so we had decided not to include the summerization of the technique in our manuscript. However, to reflect the reviewer’s suggestion, we added the paragraph describing the importance of the solution blow spinning process with citing the important articles that the reviewer suggested as below: (Line 208)

“Solution blow spinning which uses a concentric nozzles and compressed gas also has attracted the attention in nanofiber fields.[50-55] The high pressure gas such as air, nitrogen, and argon is released from the outer nozzle, and the polymer solution is ejected through the inner nozzle. This technique also has been considered as an effective way to achieve higher production rate than that of typical electrospinning.[55] This process is kindred to melt-blown technique. The polymer solution is stretched by the co-flowing gas jet from outer jet, leading to formation of a polymer nanofiber structure. Similar to blow spinning, bubble spinning (which called as foam electrospinning) also use a gas. However, the gas is injected with the submerged nozzles in solution making bubble, leading the possibility of fibers initiating from bubble surface. This type of spinning using single or several bubbles in a vast bath of solution has been shown in various reports.[56-58]”

  1. Oliveira, J.E.; Moraes, E.A.; Costa, R.G.F.; Afonso, A.S.; Mattoso, L.H.C.; Orts, W.J.; Medeiros, E.S. Nano and submicrometric fibers of poly(D,L-lactide) obtained by solution blow spinning: Process and solution variables. Journal of Applied Polymer Science 2011, 122, 3396-3405.
  2. Farias, R.M.d.C.; Menezes, R.R.; Oliveira, J.E.; de Medeiros, E.S. Production of submicrometric fibers of mullite by solution blow spinning (SBS). Materials Letters 2015, 149, 47-49.
  3. Sinha-Ray, S.; Zhang, Y.; Yarin, A.L.; Davis, S.C.; Pourdeyhimi, B. Solution Blowing of Soy Protein Fibers. Biomacromolecules 2011, 12, 2357-2363.
  4. Sinha-Ray, S.; Yarin, A.L.; Pourdeyhimi, B. The production of 100/400nm inner/outer diameter carbon tubes by solution blowing and carbonization of core–shell nanofibers. Carbon 2010, 48, 3575-3578.
  5. Sinha-Ray, S.; Sinha-Ray, S.; Yarin, A.L.; Pourdeyhimi, B. Theoretical and experimental investigation of physical mechanisms responsible for polymer nanofiber formation in solution blowing. Polymer 2015, 56, 452-463.
  6. Medeiros, E.S.; Glenn, G.M.; Klamczynski, A.P.; Orts, W.J.; Mattoso, L.H.C. Solution blow spinning: A new method to produce micro- and nanofibers from polymer solutions. Journal of Applied Polymer Science 2009, 113, 2322-2330.
  7. Yong Liu; Ji-Huan He; Lan Xu; Jian-Yong Yu. The principle of bubble electrospinning and its experimental verification. 2008, 28, 55.
  8. Y. Liu; J.-H. He. Bubble Electrospinning for Mass Production of Nanofibers. 2007, 8, 393.
  9. Liu, Y.; Ren, Z.F.; He, J.H. Bubble electrospinning method for preparation of aligned nanofibre mat. Materials Science and Technology 2010, 26, 1309-1312.

Round 2

Reviewer 2 Report

The revision is satisfying. 

Reviewer 3 Report

Accept in its present format